# Benefits of Legume Species in an Agroforestry Production System of Yellow Pitahaya in the Ecuadorian Amazon

Yadira Vargas-Tierras [1], Alejandra Díaz [1], Carlos Caicedo [1], Julio Macas [1], Alfonso Suárez-Tapia [2,*] and William Viera [3]

1 National Institute of Agricultural Research (INIAP), Central Amazon Research Site (EECA), Joya de los Sachas 220350, Ecuador; yadira.vargas@iniap.gob.ec (Y.V.-T.); alejandra.diaz@iniap.gob.ec (A.D.); carlos.caicedo@iniap.gob.ec (C.C.); julio.macas@iniap.gob.ec (J.M.)
2 Graduate School of Agroindustry and Food Science, Campus Queri, Universidad de las Américas (UDLA), Quito 170513, Ecuador
3 National Institute of Agricultural Research (INIAP), Santa Catalina Research Site, Tumbaco Experimental Farm, Tumbaco 170902, Ecuador; william.viera@iniap.gob.ec
* Correspondence: alfonso.suarez@udla.edu.ec; Tel.: +593-996-759-124

**Abstract:** Agroforestry systems have become an alternative that promotes the conservation of natural resources and the sustainable production of fruit crops in the Ecuadorian Amazon. However, it is required to demonstrate the benefit of the companion species that make up these production systems. The objective of this research was to determine how the legume species within an agroforestry system influence the yield of yellow dragon fruit (pitahaya), carbon sequestration and nutritional contribution. The experiment was carried out in Palora (province of Morona Santiago) and organized in a randomized complete block design with three replications. The treatments were two agroforestry arrangements and the monoculture as a control treatment. *Erythrina poeppigiana*, *Gliricidia sepium* and *Flemingia macrophylla* were used in the agroforestry arrangements for the contribution of biomass. Results showed that during the five years of study, pitahaya yield was influenced by the quality of the leaf litter (biomass) incorporated in to the fruit crop. Biomass from *E. poeppigiana* and *F. macrophylla* as companion crops contributed a greater amount of Ca and Mg, increased C sequestration and crop yield. The results suggest that the use of legume species in agroforestry systems positively affects pitahaya productivity, enabling sustainable agriculture in the Ecuadorian Amazon.

**Keywords:** sistema agroforestal; legumbres; cultivo de frutas; carbón

## 1. Introduction

Yellow pitahaya (*Hylocereus megalanthus* sinonimus of *Selenicereus megalanthus*) is an exotic fruit that is desired worldwide for its flavor, appearance and quality, it possesses nutritional and bioactive components (glucose content, betalains, vitamins, organic acids, dietary soluble fiber, phytoalbumins and minerals) that has allowed the fruit to be considered as a functional food [1,2]. This has created an increase demand of this fruit, which has resulted in the expansion of the cultivation area in many countries such as USA, Mexico, Guatemala, El Salvador, Nicaragua, Costa Rica, Venezuela, Panama, Uruguay, Peru, Brazil, Ecuador, Colombia, Thailand, Indonesia and Vietnam [3].

In Ecuador, there are about 850 hectares of pitahaya [4]; however, in recent years the cultivation area has been steadily increasing as exports have grown. In 2019, 7498.80 tons of fruit were exported to Hong Kong, USA, Russia, the Netherlands, France, Germany and Spain, generating more than US $44 million in income for the country [5]. In the country, it is grown mainly in the provinces of Pichincha, Manabí and in the Ecuadorian Amazon in Morona Santiago, Orellana and Sucumbíos [6,7].

Currently, pitahaya is grown commercially as a monoculture with conventional agronomic management (high use of agrochemicals), a production technology that has caused

negative impacts on natural resources, such as loss of biodiversity, soil degradation and erosion due to the excessive use of agrochemicals, and total destruction of the ecosystem [8,9]. This type of agricultural practices worldwide is causing extreme fluctuations in climatic conditions, such as temperature increase, alteration of rainfall distribution, droughts or floods caused by the concentration of greenhouse gases $CO_2$, methane and nitrous oxides in the atmosphere [10,11]; conditions that threaten food security [12]. New production alternatives are being investigated to reduce greenhouse gas emissions in order to mitigate the most extreme effects of climate change [13].

Among these alternatives, there are the agroforestry systems (AFS) that has the potential to face both food security and carbon (C) sequestration mitigation goals [14,15]. These systems combine commercial crops (fruit trees), shrubs and trees [6,16] in the same area to improve the economic gains of the producers [17]. In addition, it has been identified as a potential production approach for greenhouse gas mitigation under the Kyoto Protocol [13].

Several studies have determined that AFSs store more carbon than monocultures, depending on biological, climatic, edaphic and site-specific management conditions [17–19]. Ref. [20] reported that an AFS stored twice as much carbon as a monoculture (34.61 t C ha$^{-1}$ and 18.74 t C ha$^{-1}$, respectively). This same behavior was found in the Ecuadorian Amazon in chakra-type systems (a traditional and diverse AFS) [7] with seed-propagated cacao, where C storage was 141.4 t ha$^{-1}$ and in a monoculture was from 4.9 to 7.6 t ha$^{-1}$ [21]. In other studies conducted in AFS in the Peruvian and Colombian Amazon, Panama and Costa Rica; the amount of C stored in cocoa AFSs with scattered trees (timber and fruit trees) was 131.18 (65.61 for aerial biomass and 65.57 for the soil component), 61, 43 to 60 and 50 to 100 t ha$^{-1}$ per year, respectively. In Mexico and Costa Rica, in coffee with *Erythrina poeppigiana*, an accumulation of 115 and 195 t ha$^{-1}$ of C per year has been reported; and in Filipinas it was 93 t ha$^{-1}$ per year in cocoa with *Gliricidia sepium* [22–26]. In terms of economic approach, it has been reported that the producers' income improves 13 times more in systems with fruit species (*Theobroma cacao*, *Cocos nucifera* and *Coffea* sp.) and short cycle species (*Zea mays* and *Oryza sativa*) [27]. In addition, [28] points out that agroforestry systems are 36 to 100% more productive than monocultures, depending on the type of crop, crop arrangement and edaphoclimatic conditions.

Studies of C sequestration in AFSs with fruit trees are scarce, and specifically none or non-existent with the yellow pitahaya crop. Considering this situation, this study was initiated to investigate the hypothesis that pitahaya planted with forest species does not decrease its yield and that leguminous forest species provide nutrients to the crop and store a significant amount of carbon. The main objective of this study was to estimate the agronomic behavior of pitahaya crop in agroforestry systems compared to the monoculture. To reach this purpose, pitahaya yield, carbon storage and nutritional contribution by leguminous species was evaluated in the agroforestry system.

## 2. Materials and Methods

### 2.1. Location of the Experiment

This study was conducted from 2016 to 2020 at the Palora Experimental Farm of the Central Experimental Station of the Amazon (EECA) of the National Institute of Agricultural Research (INIAP), located in the canton Palora, province of Morona Santiago. The experimental site was located at 1°40'14.5'' S latitude and 77°57'50.3'' W longitude (Figure 1), with an altitude of 864 masl, corresponding to the Piedmont Periandine [29]. The climate of the study area is humid subtropical, with an average temperature of 20 °C, relative humidity of 89% and rainfall of 3122 mm year$^{-1}$.

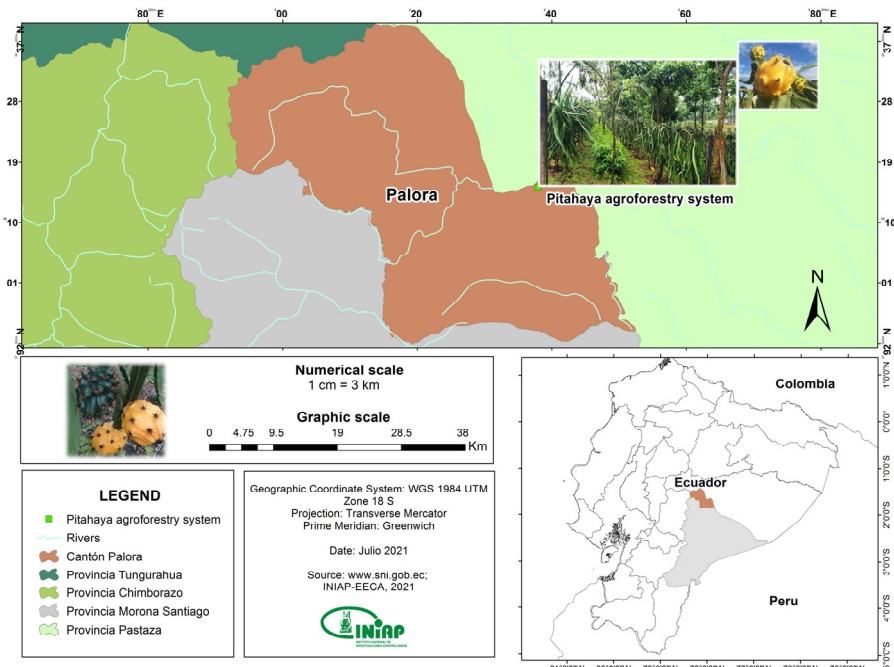

**Figure 1.** Location of the pitahaya agroforestry system assay in the county of Palora, province of Morona Santiago.

According to the Development and Territorial Planning Plan for the year 2015–2025, Palora is located at 920 masl, in the northwest of the province of Morona Santiago. It has 6936 inhabitants, where 80% live in the urban area and 20% in the rural sector, with an annual population growth rate of 1.04%. The 46% of the population belongs to the economically active population which is dedicated to agriculture, livestock, forestry and fishing; while the rest of the population is dedicated to other activities, such as commerce wholesale and retail, public and private administration, construction, education, manufacturing and transportation industries. Many years ago, farmers were mainly dedicated to the cultivation of *Bixa orellana*, *Carica papaya*, *Manihot esculenta*, *Musa paradisiaca*, *Colocasia esculenta*, *Solanum quitoense*, *Saccharum officinarum*, *Camellia sinensis* and citrus fruits. However, in recent years, agriculture is based on the production of a non-traditional fruit crops such as pitahaya which provides a relatively high profit margin to producers.

In 2020, the National Planning Secretary of Ecuador points out that 77% of the fruit produced in Palora is marketed nationally and exported to Hong Kong, Asia, the United States, Russia, the Netherlands, France, Germany and Spain which are the main consumers of pitahaya in the world [6,30] (Vargas, 2020). Therefore, this fruit crop has generated sources of employment improving the quality of life of at least 80% of the population (National Planning Secretary, 2020). Due the growth in the demand for this fruit and the fact that producers needed to organize, the "Association of Producers and Marketing of Pitahaya and other Palora products" was created, which is currently made up of 672 active members [6] (Vargas, 2020).

### 2.2. Experimental Treatments

The experiment was organized in a randomized complete block design with three replications, the treatments being two agroforestry arrangements and monoculture as a control. For the agroforestry arrangements, multipurpose trees of *Erythrina poeppigiana* and *Gliricidia sepium* were used, each with the yellow pitahaya crop. Additionally, a shrub legume (*Flemingia macrophylla*) was planted in the two agroforestry arrangements to provide biomass. The three species used in the agroforestry arrangements are considered to have high agronomic potential as nitrogen (N) fixers and soil improvers (structure) [31]. Multipurpose trees have two functions: providing shade to the crop and the extraction

of nutrients from deeper soil layers that are then deposited on the surface in the form of organic residues (biomass) resulting from frequent pruning of these trees [32]. *F. macrophylla* was used because producers in the northern Amazon use it as a cover species to conserve and protect the soil [33] and conserve macrofauna [34].

### 2.3. Crop Management

At the beginning of the trial, the yellow pitahaya plants were two years old. They were planted at 3 m between rows and 2.5 m between plants, using inert concrete stakes 1.80 m high, two posts per plant with two lines of galvanized wire number 10 at 2.5 m. The multipurpose trees (*E. poeppigiana* and *G. sepium*) were transplanted 6 m between rows and 6 m between plants and the arbustiva species (*F. macrophylla*) were planted in double rows in the rows where the multipurpose species were located, with a spacing of 0.50 m between rows × 0.50 m between plants. The gross and net plot sizes were 15.0 × 25 m and 6.0 × 8.0 m, respectively. The net plot area was used to determine the crop yield.

Six months after planting, the basal shoots of the multipurpose trees were removed in order to leave only a single stem. In the first year, the multipurpose trees were pruned, eliminating lower branches and forming the canopy from 4 m [35]. *F. macrophylla* were not pruned.

Shade species biomass cutting consisted of eliminating 60% of the aerial biomass. The amount of biomass incorporated from *E. poeppigiana* ranged from 26 to 50 kg plant$^{-1}$ between the first year and the fifth year of biomass incorporation; while that of *G. sepium* ranged from 29 to 10 kg plant$^{-1}$ between the first year and the fifth year. Biomass incorporation was carried out every 120 days (3 pruning per year). In contrast *F. macrophylla* plants were pruned every 90 days (4 pruning per year) when they presented 50% flowering, adding 1 to 4 kg plant$^{-1}$ between the first year and the fifth year of biomass incorporation [36,37]; this pruning was done 20 cm above the ground. All organic matter was chopped and left on the soil surface next to the pitahaya plants, as recommended by [38].

From the second year on, two sanitary pruning were carried out in the pitahaya crop in April and December; removing the unproductive branches, diseased stalks and intertwined branches, allowing good air circulation, reducing the weight of the plants on the training system and reducing the spread of pathogens [6].

The amount of nutrients applied was calculated according to the procedure described by [6], which considers the crop requirement, the contribution of nutrients from the soil according to its fertility (Table 1), and the efficiency of the fertilizer.

**Table 1.** Soil chemical characteristics in the five years of evaluation.

| Description | pH | ppm | | | (meq/100 mL) | | | % |
|:---:|:---:|:---:|:---:|:---:|:---:|:---:|:---:|:---:|
| | | N | P | S | K | Ca | Mg | M.O |
| Year 1 | 5.4 | 38.0 M | 12.7 M | 6.40 L | 0.1 L | 2.5 L | 0.4 L | 13.7 H |
| Year 2 | 5.4 | 51.1 H | 6.00 L | 5.80 L | 0.1 L | 1.2 L | 0.4 L | 12.5 H |
| Year 3 | 5.6 | 34.2 M | 25.6 H | 20.4 M | 0.3 M | 6.9 M | 0.8 L | 14.3 H |
| Year 4 | 5.2 | 47.0 H | 7.70 L | 3.40 L | 0.1 L | 2.7 L | 0.5 L | 15.8 H |
| Year 5 | 5.4 | 56.9 H | 19.3 M | 7.50 L | 0.2 L | 7.5 M | 0.6 L | 15.6 H |

H: High; M: Medium; L: Low.

The fertilizers used in this experiment were ammonium nitrate (38% N), di-ammonium phosphate (18% N, 48% P), magnesium sulfate (12% Mg, 20% S), and potassium chloride (60% K). A total of 369 to 177 g N plant$^{-1}$, 322 to 120 g P plant$^{-1}$, 480 to 292 g K plant$^{-1}$ and 162 to 300 g S plant$^{-1}$ were applied, the highest amount of these elements being applied in year one and decreasing in year five, respectively.

In the case of N, it was applied four times a year, 50% was applied in the vegetative growth stage (June and August), and another 50% in the reproductive stage (September and December). In the vegetative stage, 40% of P was applied and two fractions (30% each) were applied in the reproductive stage. K was applied in the vegetative stage 20% and

two fractions in the reproductive stage (60% and 20%). Sulfur was applied 33% in the vegetative stage and 67% in the reproductive stage.

Phytosanitary controls were carried out every 15 days, using preventive and curative agrochemicals such as chlorothalonil, cymoxanil, metalaxyl and mancozeb, abamectin and lambdacyhalothrin. Weeding was carried out monthly with a brush cutter. Fruit was harvested manually with pruning shears when the fruit was at ripening stage 4 (Figure 2).

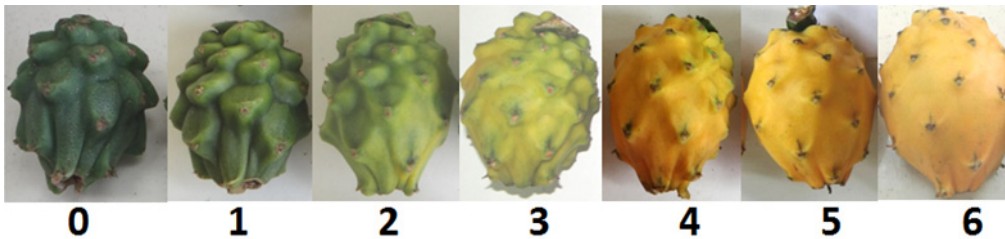

**Figure 2.** Pitahaya fruit maturity grades. Note the color change, the fruits were harvested at grade number 4.

### 2.4. Plant Sampling and Analysis

Fresh biomass of legumes. Three shade trees (*E. poeppigiana* and *G. sepium*) were taken from the net plot to quantify the biomass contribution of the pruning. In situ, the total pruning biomass (leaves and branches) was weighed in kg using a balance. To determine the amount of fresh biomass per hectare, the average biomass was determined and multiplied by the number of trees. To quantify the total biomass of pruned shrub legumes (*F. macrophylla*), three rows (232 plants) were taken from the total plot, the average biomass per row was determined and multiplied by the number of rows per hectare [32]. On the other hand, the biomass of the shade and shrub species was added to determine the total amount of biomass per hectare per year for each agroforestry system (treatments).

Composite samples (leaves and branches) of 250 g were taken from each pruning, deposited in paper bags duly identified and taken to the Soil and Water laboratory of the EECA for dry matter, N, P, K, Ca, Mg and S determination analysis.

Concentration of nutrients in the biomass. For the calculation of nutrient contents, the total biomass produced by each treatment was multiplied by the dry matter produced by each legume species and then the equation for macroelements suggested by [36] was applied.

$$Q = [MST \times X]/10^2 \tag{1}$$

Q = Nutrient content in total dry matter (expressed in kg of nutrient ha$^{-1}$)
MST = Total dry matter
X = Concentration of nutrient in dry matter

The nutrient content obtained in the treatments and replications was extrapolated to estimate the contribution of N, P, K, Ca, Mg and S in kg per hectare per year. For the determination of total N, the Semimicro Kjeldahl method was used, P was determined by the nitric-perchloric digestion extract colorimetric method, while K, Ca, Mg and S were determined by atomic absorption spectrometry [39].

For the determination of total organic carbon of the system components (multipurpose legumes and shrubs), C present in the biomass was assumed to be 50% [40,41]. The C stocks obtained in each treatment were extrapolated to estimate the stocks per hectare.

**Soil nutrient concentration**. Once a year, composite samples of 1 kg (per treatment and repetition) were taken and deposited in plastic bags properly identified and taken to the Soil and Water laboratory of the EECA to carry out the analyses of organic C, total N, P, K, Ca, Mg and S. The determination of N was carried out with the Semimicro Kjeldahl method. The determination of P, K, Ca, Mg and S was carried out from an extract obtained by the modified Olsen method (pH 8.5) and P was determined by colorimetry, K, Ca and Mg by atomic absorption spectrometry and S by turbidimetry. C was determined by

the Walkey and Black method [39]. With the C and N values, the carbon/nitrogen ratio was determined.

**Pitahaya fruit yield.** Crop yield was determined in the net plot by weighing all fruits (g plant$^{-1}$). The number of fruits per plant and by size (size 20 = fruits $\leq$ 110 g, size 14 = 111 to 150 g, size 2 = 151 to 250 g, size 9 > 250 g) was counted [42]. The yield obtained was extrapolated per hectare.

### 2.5. Data Analysis

### 2.5.1. Multivariate Statistics

A multivariate analysis was performed using the R (3.2.4) program coupled with the MDA tools package (doi: 10.5281/zenodo.59547). A Principal component analysis model was used to identify the companion crops that affect pitahaya yield. The companion crops were: *E. poeppigiana + F. macrophylla*, *G. sepium + F. macrophylla* and monoculture was set as control. This experiment was carried out from 2016 to 2020. Scaling was performed because the variables were expressed in different units.

### 2.5.2. Response Analysis

The responses identified by PCA due to different years and complaining crops were modeled as:

$$Y_{ijkl} = \mu + A_i + B_j + T_k + AT_{ik} + \varepsilon_{(ijk)l} \tag{2}$$

4 Years = i = (2016, 2017, 2018, 2019, 2020) 3 blocks j = 1, 2, 3 3 Companion crops k = 1 to 3, where:

$Y_{ijkl}$ = denotes the observation at the ith year, the jth block for the kth companion crop
$\mu$ = grand mean
$A_i$ = random effect of the ith year
$B_j$ = random effect of the jth block
$T_k$ = fixed effect of the kth companion crop
$AT_{ik}$ = interaction effect between the ith year and the kth companion crop
$\varepsilon_{(ijk)l}$ = Random experimental error (0, $\sigma_e^2$).

The data was analyzed using SAS 9.4 mixed model procedure. A repeated measures approach was used for successive sampling years, using a variance-covariance structure which was selected based on the lowest Akaike's Information Criterion [43].

The data was analyzed using SAS 9.3 mixed model procedure. Interactions and main effects were declared significant at $p < 0.1$. Variables were checked for assumptions of normality and homogeneity of variances based on plot of residuals vs. predicted values. Transformations were performed as needed to comply with the normality assumption. The transformations were based on the Box-Cox power transformation series [44]. Least square means were separated using Tukey mean separation (alpha = 0.1) procedure in SAS proc mixed and the means were order using mean separation into groups by letters [45].

### 3. Results

*PCA Modeling*

The PCA model (Figure 3) shows that yield is related with higher number of fruits, higher amount of Ca and Mg in biomass and the lowest C:N ratio in the biomass. Biomass arises from the companion crops, therefore in 2016 we did not have enough biomass for sampling. The PCA shows that during the five years of this experiment, the treatment *E. poeppigiana + F. macrophylla* provided higher Ca and Mg in the biomass and the lower C:N compared with the other companion crop. Moreover, the yield appears to be influenced by the quality of the companion crop.

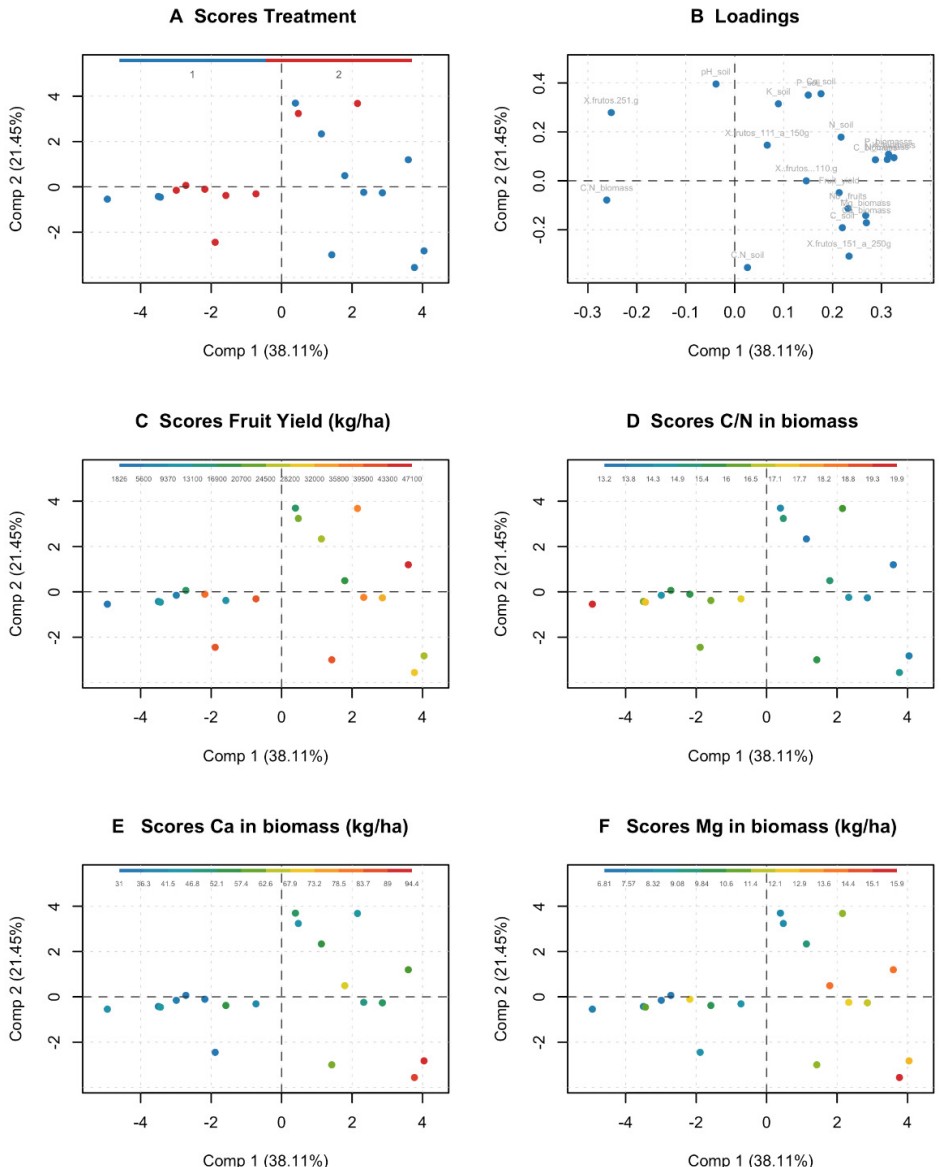

**Figure 3.** Principal component analysis on pitahaya yield. Panel (**B**) is the loading plot and panels (**A**–**F**) the score plots. The principal component 1 and 2 are plotted in panel (**A,B**) showing the influence of the companion crops over yield from 2017 to 2020, data from 2016 and monoculture excluded since there was no biomass for sampling. Panels (**C**–**F**) provide a representation of the score plot colored with respect to the amount of a given variable (from low to high). Treatments 1 to 3 (panel (**A**)) are (1) *E. poeppigiana* + *F. macrophylla*, (2) *G. sepium* + *F. macrophylla,* respectively.

Calcium in biomass (Figure 3) shows that ranges > 55 kg ha$^{-1}$ and in Magnesium > 12 kg ha$^{-1}$ are related with higher number of fruit and yield. The C:N ratio in the biomass shows as well that ranges > 15 are related with higher yield, mostly coming from more biomass from the *E. poeppigiana* + *F. macrophylla* companion crop.

A univariate analysis was performed for yield. Table 2 shows the significance of main effects and interactions. The analysis showed that there was a highly significant effect in years ($p < 0.0001$) and the agroforestry systems ($p = 0.0312$). There was no interaction between the years of evaluation and the agroforestry systems ($p = 0.3836$).

**Table 2.** Main effects and interaction effect on fruit yield determined for each factor: Year and Agroforestry systems. Mean values are reported. Within a column and within a given factor, means followed by the same letter are not statistically different (alpha = 0.1).

| Treatment | Yield (t ha$^{-1}$) |
|---|---|
| Year | ** |
| Agroforestry system | ** |
| Year x Agroforestry system | NS |

NS not significant; ** significant at $p \leq 0.01$.

The main effect for the factor years showed that the pitahaya yield (average of agroforestry systems treatments and monoculture) increased as the plant developed and reached its productive phase (maturity). Yield started in 2016 with 1.89 t ha$^{-1}$ and it reached the highest production in 2020 with 25.06 t ha$^{-1}$ (Table 3). It was determined that the fruit yield was higher in agroforestry systems and lower in monoculture in the five years of evaluation. In addition, the yield of the two agroforestry systems was statistically different in comparison to the monoculture yield which supports the results (Table 4).

**Table 3.** Mean values for fruit yield determined for the factor: Year. Mean values are reported. Within a column and within a given factor, means followed by the same letter are not statistically different (alpha = 0.1).

| Year | Yield (t ha$^{-1}$) |
|---|---|
| 2020 | 25.06 a |
| 2019 | 23.32 a |
| 2018 | 18.73 a |
| 2017 | 6.31 b |
| 2016 | 1.89 c |

**Table 4.** Mean values for fruit yield determined for the factor: Agroforestry Systems. Mean values are reported. Within a column and within a given factor, means followed by the same letter are not statistically different (alpha = 0.1).

| Agroforestry System | Yield (t ha$^{-1}$) |
|---|---|
| *G. sepium + F. macrophylla* | 17.17 a |
| *E. poeppigiana + F. macrophylla* | 12.89 a |
| Monoculture | 8.60 b |

In terms of C:N ratio (Table 5), the analysis showed that there was an inter-active effect between the evaluation years and the agroforestry systems ($p$ = 0.0148), a significant effect of the agroforestry systems ($p$ = 0.0055) and no statistical differences were observed for the year factor ($p$ = 0.1056). Control (monoculture) was not analyzed because there were no legumes in this system to set this variable.

**Table 5.** Main effects and interaction effect for the C:N ratio determined for each factor: Year and Agroforestry systems Mean values are reported. Within a column and within a given factor, means followed by the same letter are not statistically different (alpha = 0.1).

| Treatment | C:N Ratio |
|---|---|
| Year | NS |
| Agroforestry system | * |
| Year x Agroforestry system | * |

NS not significant; * significant at $p \leq 0.05$.

In the five years of evaluation, it was determined the main effect showed that the agroforestry system *E. poeppigiana + F. macrophylla* provided the lowest C:N ratio (14.73) (Table 6). The interaction analysis (Figure 4) indicated that the C:N ratio in 2019 was higher

in the agroforestry system *G. sepium + F. macrophylla* (16.98) and the opposite happened in the system *E. poeppigiana + F. macrophylla* (13.60). On the other hand, it was determined that the C:N ratio (average of agroforestry systems) decreased numerically from 15.80 in 2017 to 14.53 in 2020 (Table 7).

**Table 6.** Mean values for the C:N ratio determined for the factor: Agroforestry Systems. Mean values are reported. Within a column and within a given factor, means followed by the same letter are not statistically different (alpha = 0.1).

| Agroforestry System | C:N Ratio |
|---|---|
| *E. poeppigiana + F. macrophylla* | 14.73 b |
| *G. sepium + F. macrophylla* | 15.83 a |

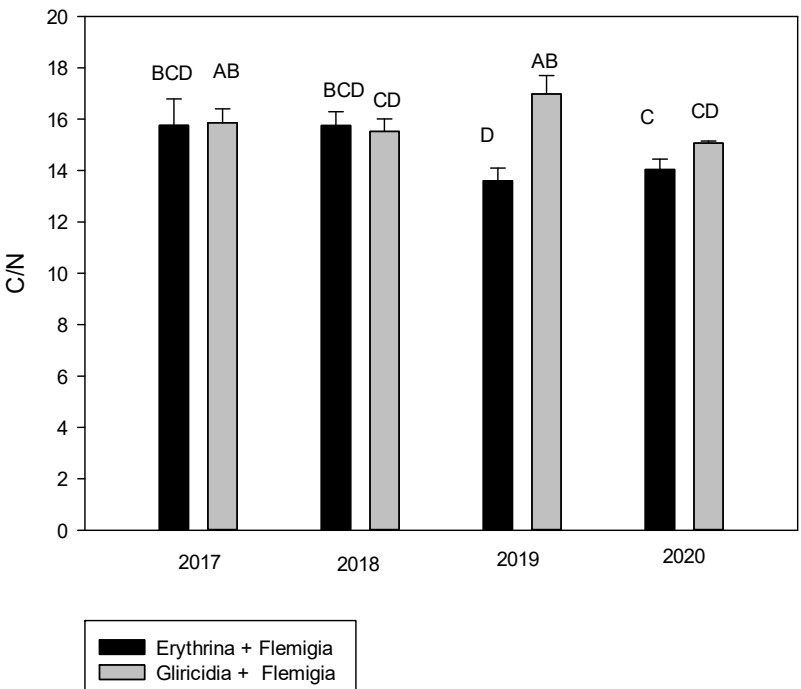

**Figure 4.** Interaction between agroforestry system and year for the C:N in biomass. In general, *G. sepium + F. macrophylla* provided the highest amount of C:N and *E. poeppigiana + F. macrophylla* the lowest. Means with different letter are significantly different at 5% level.

**Table 7.** Mean values for the C:N ratio determined for the factor: Year. Mean values are reported. Within a column and within a given factor, means followed by the same letter are not statistically different (alpha = 0.1).

| Year | C:N Ratio |
|---|---|
| 2020 | 14.53 |
| 2019 | 15.15 |
| 2018 | 15.63 |
| 2017 | 15.80 |

For the Ca content in biomass, the analysis showed that there were no statistical differences for years ($p = 0.1214$), agroforestry systems ($p = 0.1305$) and interaction year x agroforestry system ($p = 0.1206$). The analysis of the Mg content showed that there were statistical differences for the agroforestry systems ($p = 0.0114$), but not significant for years ($p = 0.6973$) and for the interaction year x agroforestry system ($p = 0.3406$) (Table 8). Control is not included in the analysis because there was no legume species that contribute with biomass.

**Table 8.** Main effects and interaction effect for Mg determined for each factor: Year and Agroforestry systems. Mean values are reported. Within a column and within a given factor, means followed by the same letter are not statistically different (alpha = 0.1).

| Treatment | Mg (kg ha$^{-1}$) |
|---|---|
| Year | NS |
| Agroforestry system | * |
| Year x Agroforestry system | NS |

NS not significant; * significant at $p \leq 0.05$.

In the five years of evaluation, the highest Ca and Mg content was obtained with the agroforestry system of *E. poeppigiana.* + *F. macrophylla* with 50.05 y 10.02 kg ha$^{-1}$, respectively (Table 9). It was also found that the contribution of Ca by legumes in the four years of evaluation is relatively variable. The Mg contribution (average of agroforestry systems) increased as the years progressed with respect to 2017 (Table 10).

**Table 9.** Mean values the Ca y Mg (kg ha$^{-1}$) determined for the factor: Agroforestry systems. Mean values are reported. Within a column and within a given factor, means followed by the same letter are not statistically different (alpha = 0.1).

| Agroforestry System | Ca (kg ha$^{-1}$) | Mg (kg ha$^{-1}$) |
|---|---|---|
| *E. poeppigiana* + *F. macrophylla* | 50.05 | 10.02 a |
| *G. sepium* + *F. macrophylla* | 33.99 | 7.46 b |

**Table 10.** Mean values for Ca and Mg (%) determined for the factor: Year. Mean values are reported. Within a column and within a given factor, means followed by the same letter are not statistically different (alpha = 0.1).

| Year | Ca (kg ha$^{-1}$) | Mg (kg ha$^{-1}$) |
|---|---|---|
| 2020 | 40.92 | 9.26 |
| 2019 | 38.67 | 10.14 |
| 2018 | 50.41 | 9.45 |
| 2017 | 36.28 | 8.13 |

## 4. Discussion

Pitahaya yield increased from 1.89 t ha$^{-1}$ in 2016 to 25.06 t ha$^{-1}$ in 2020. The yield variation in the different years of evaluation is possibly due to the fact that the plant is reaching its full maturity (fruiting age), similar yields have been reported by [46] in pitahaya monocultures where in the first two years a plant of pitahaya produces 1.0 t ha$^{-1}$ and between the fifth and sixth years, the yield stabilizes at 18 kg per plant.

In addition, it was determined that the highest yield was obtained in the two agroforestry systems with 12.89 t ha$^{-1}$ (*E. poeppigiana* + *F. macrophylla*) and 17.17 t ha$^{-1}$ (*G. sepium* + *F. macrophylla*) in relation to the monoculture (8.60 t ha$^{-1}$). Yields obtained in agroforestry arrangements differ from those reported by [3] for sustainable pitahaya systems with live *Bursera simaruba* tutors where in third year reached 20 t ha$^{-1}$ and 17 t ha$^{-1}$ for a monoculture.

This same favorable behavior for yield increase with the agroforestry systems, was obtained in an intercropped system of *Zea mays* and *Sorghum* sp. with *G. sepium*, where yield was higher by 42 and 55%, respectively, compared to the monoculture; in addition the yield stabilized over time as observed in our study with pitahaya [47]. This trend has also been reported in agroforestry systems of coffee with legume shade (*G. sepium* + *Leucaena leucocephala* + *E. variegata*), where yields increased by 20% [46,48] also points out that it is very difficult to estimate fruit yield of pitahaya because it depends a lot on the age of the crop, agronomic management, production systems, planting distance, climate and sexual incompatibility.

In 2016 (first year), the number of fruits per plant were 9 and 11 in the agroforestry systems *E. poeppigiana + F. macrophylla* and *G. sepium + F. macrophylla*, respectively, and 2 in the monoculture, While in 2020, it increased to 106 in the *E. poeppigiana + F. macrophylla* system and 56 in the *G. sepium + F. macrophylla* system but it was lower in the monoculture (38). The relationship between number of fruits and yield is mainly due to the fact that this variable is an important component of yield [49].

In contrast the soluble solids content of the pitahaya fruits from the agroforestry systems and the monoculture was determined in 2020. A value of 21.1° Brix was obtained in the *E. poeppigiana + F. macrophylla* system, whereas the *G. sepium + F. macrophylla* system reached 21.4° Brix; on the another hand, the monoculture got 22° Brix; these values are similar among them thus this variable does not indicate any significant difference. These results were relatively higher than the values reported by [1], who mentions that the yellow pitahaya harvested at maturity stage 6 in the Palora site was 20.74° Brix.

In agroforestry systems, the strong relationship between yield and nutrients provided by biomass (Ca and Mg) is possibly due to the fact that these elements are absorbed by the deep roots of the trees and deposited through the litter on the most superficial soils, that is, the effect of nutrient pumping occurs [50]. This direct supply of nutrients from decomposition processes and mulching effects improve crop yield [51].

Calcium (>39 kg ha$^{-1}$) and Mg (>10 kg ha$^{-1}$) contained in the biomass of the legumes species used in the agroforestry systems are directly related to crop yield. This is possibly due to the fact that these species store high concentrations of N, P, K, Ca and Mg in their leaves; which can be supported by [52], who reports that *Tithonia diversifolia* accumulates high concentrations of nutrients in its leaves and that the moment they are incorporated into the soil, they improve the nutritional status through the mobilization and return of nutrients, as well as to the contribution of organic matter [53]. Ca and Mg are the most important mineral nutrients in production [54,55]. A higher yield, larger fruit size, higher percentage of marketable fruit [54] and better shelf life in crops are achieved when Ca is present in crop nutrition [55]. According to the results, Mg influenced it to improve pitahaya yield in agroforestry systems by 29% with respect to monoculture [56] who points out that Mg improves fruit yield by 12.5% when fruit crops grown in acidic soils that was the case of this study (Palora soil pH 5.2 to 5.6).

In addition, it was determined that the Ca and Mg content varied from one year to another and that the highest contribution was achieved with the agroforestry system *E. poeppigiana + F. macrophylla* which contributed with amount of biomass of 19,753 kg ha$^{-1}$, while the *G. sepium + F. macrophylla* system contributed with 6700 kg ha$^{-1}$, consequently the nutrient content basically depends on the amount of biomass produced by the leguminous species for the system. This result is corroborated by [51] who mentions that the amount of nutrients that is incorporated depends on the amount of biomass that is produced with pruning. For example, [57] in an agroforestry system study, determined that although *G. sepium* contains the highest concentration of nutrients in its leaves, it did not produce abundant biomass like *L. leucocephala*, causing this legume to be considered the best source of nutrients because it incorporated 144 kg ha$^{-1}$ of Ca and 60 kg ha$^{-1}$ of Mg, amounts that would meet the nutritional needs of various crops.

The association of yield with the C:N ratio obtained in this study is possibly due to the fact that legumes produce a lower C:N ratio than tree and grass species in agroforestry systems [58], which would favor the release of N to the soil, an effect that is reflected in the crop yield. This is corroborated by [59], who mentions that legume species contribute to natural regeneration due to their association with N-fixing bacteria, helping to increase the yield and fertility of agroforestry soils by producing a high quality leaf litter. Ref. [60] indicated that an agroforestry system of *Calliandra calothyrsus* and *Sesbania sesban* with *Zea mays*, these species contributed more N and competed favorably with associated crops than when agricultural crops were with non-legume species.

It was determined that the greater contribution of biomass in the agroforestry system of *E. poeppigiana + F. macrophylla* increased the yield when the C:N ratio was around 15.

It was supported by [61], who points out that when the C:N ratio is <20 or 30 there is a mineralization of N and when the ratio is >30 there is a immobilization of N. In addition, it was determined that the C:N ratio in the years of evaluation was decreasing and the lowest ratio was achieved with the *E. poeppigiana + F. macrophylla* system in 2020. Ref. [58] points out that the decomposition rate should not be determined only by the C:N ratio but also by other characteristics of the plant such as chemical composition, size, distribution, branching of the shoots and roots, as well as the texture of the soil, temporal variations of the gas content and water with time and the atmosphere (evapotranspiration rates for each type of plant over), thus the activity of a series of microorganisms in the soil [55]. Therefore, the presence of legumes in agroforestry systems improves the quality of litter, due to the fact that their residues rich in N improve the efficiency of the use of C by part of the decomposing microorganisms, which results in a higher microbial biomass [62] and the eventual stabilization of soil organic C [59].

In previous studies in coffee agroforestry systems with *E. poeppigiana*, it was determined that 50% N is fixed by this legume and it is a species that is recommended to be able to substitute the use of nitrogen fertilizers [63]. In addition, it is indicated that the mineralization processes of P, K, Ca and Mg of *E. poeppigiana* leaves are faster than in other species such as *Inga edulis* or *Cajanus cajan*, approximately 40% of the initial P and Ca contents and the 75% of the Mg and K content of the leaves of *E. poeppigiana* are mineralized in four weeks [64] which is advantageous for the nutrition of a crop in association with this species.

Finally, this kind of sustainable production systems will strengthen the pitahaya production to encourage integrated fruit production in transition to organic fruit production which can be mainly for exportation [65]. On the other hand, in the Ecuadorian Amazon, these sustainable agriculture alternatives (agroforestry systems) help improve soil health by reducing erosion rates by 50%, increasing the infiltration rate by 75%, reducing runoff by 35%, increasing soil organic C by 21%, storing organic N by 13% and available N by 46%, and phosphorous is available by 11%. In addition, these systems contribute to reduce the use of external inputs (fertilizers) because the associated species incorporate nutrients into the system, improve crop profitability because the income comes from different sources, guarantee food security because 40% of the production is used for family nutrition, conservation and local use of the agrobiodiversity and provide ecosystem services (regulation of hydrological cycles, recovery of degraded and contaminated areas) [30] The legumes present in agroforestry systems improve crop yield, reduce production costs by 30% due to the reduction of the use of external inputs such as synthetic fertilizers and improve producers' income.

## 5. Conclusions

Agroforestry systems produced enough biomass and nutrients that helped meet the demand for pitahaya cultivation. The patterns of the C:N ratio, Ca and Mg content (biomass) in the different years of evaluation allowed to determine that the benefit of the biomass added through pruning is not immediate but long-term.

In the agroforestry systems, the amount of nutrients that were incorporated into the crop depended on the amount of biomass that was added through pruning, an event that allowed to infer that the agrosystem *E. poeppigiana + F. macrophylla* could be considered as the best source of nutrients for yellow pitahaya grown in the Palora site because they incorporated 50.05 kg ha$^{-1}$ of Ca and 10.02 kg ha$^{-1}$ of Mg, quantities that meet the nutritional needs of this fruit crop. In addition, it reduces production costs by 30% due to the reduction in the use of external inputs (fertilizers).

**Author Contributions:** Conceptualization, Y.V.-T. and C.C.; methodology, Y.V.-T., C.C. and A.D.; statistical analysis, A.S.-T., W.V., writing—original draft preparation, W.V., Y.V.-T., A.D. and J.M.; writing—review and editing, W.V., Y.V.-T., A.S.-T. and A.D. All authors have read and agreed to the published version of the manuscript.

**Funding:** This research was financed by the National Institute of Agricultural Research (INIAP), through the Central Amazon Research Site (EECA).

**Institutional Review Board Statement:** Not applicable.

**Informed Consent Statement:** Not applicable.

**Data Availability Statement:** Data is contained within the article.

**Acknowledgments:** We thank the agronomists of the EECA Fruit Program, Tissa Kannangara for editing the manuscript and the reviewers for their valuable comments and suggestions on this document.

**Conflicts of Interest:** The authors declare no conflict of interest.

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
