# Peer review of "Benefits of Legume Species in an Agroforestry Production System of Yellow Pitahaya in the Ecuadorian Amazon"

_sustainability, doi:10.3390/su13169261_

Round 1

Reviewer 1 Report

First, I would like to congratulate the authors to work on such a nice system and I approve their effort to improve current agricultural methods.

The manuscript is well written and clear. I agree with most of the results and the conclusions drawn in the manuscript. The discussion is very interesting and add a lot to the story.

However, there is one major issue concerning the main analysis of the results. The PCA preformed and displayed in Fig1 does not seems to be right.

Indeed, many data are only possible to be measured in your 2 agroforestry systems and not in the monoculture condition (ie: Ca, C/N amount in biomass, etc...). Hence it influences drastically the PCA analysis, toward agroforestry as they have much more data in the PCA. Then this approach is not statistically valid. At least not without the appropriate correction for missing data. These corrections are usually difficult to evaluate.

Another PCA should be performed using only the data that are present/measured in the 3 treatments.

Data only measured in 2 treatments can be analyzed separately as already done in the manuscript. Or another PCA including only the 2 treatments can be performed too.

Other issues:

Fig1: 

Titles need to be re-formatted.

Place the letter first (A, B etc..) then the titles, add the units used instead of score. What does score Ca in biomass mean? is it only the Ca amount in biomass or the % as in Table 6?

Table 3, years not “anos”  

These tables are not very clear (Table 3, 5, 6, 8). The yield per year is calculated for each year, but is it an average of every conditions? or only one condition ?

For example in the table “G. sepium + F. macrophylla 17.17”  is the average yield per year or something else ?

Please explain a bit more how these table are made and how the numbers are calculated. Even better remake these table to be easily understandable.

L284 maturity

Reformulate L285

the highest yields were obtained with the two agroforestry arrangements that were statistically similar and different from monoculture (Table 3).

I don’t understand this sentence.

Fig3: error bar representing the standard deviation are missing.

Author Response

 "Consulte el archivo adjunto."

Reviewer 2 Report

In the introduction, it would be necessary to review studies that present other models of good practice in terms of the economic benefits.

In the methodology, a map should be introduced with the exact location of the studied area in relation to the human settlements and landforms, precisely so that the reader can locate the production area in relation to the physical-geographical factors or human potential of the region. It is mandatory for the authors to specify the purpose of the study and the objectives pursued.

Also, the economic and demographic specifics of the studied area should be presented: what types of agricultural activities were practiced in the past, what is the current specifics, what is the evolution of the population in the area, what is the degree of employment in agricultural activities, if there is another type of economy, etc.

The results present in detail the relationship between biomass and agroforestry system, but it would be necessary to explain the relationship between benefits and the local economy. Also, in order for the results to be improved, it would be necessary for the authors to insist on the advantages or disadvantages (if any) manifested on the population in the area, which appeared with the implementation of this agricultural system. The idea of economic benefits for the population should also be discussed in the discussion and conclusion chapters.

Round 2

Reviewer 2 Report

I agree with all the changes proposed by the authors.